# Global Vibration Comfort Evaluation of Footbridges Based on Computer Vision

**DOI:** 10.3390/s22187077

**Published:** 2022-09-19

**Authors:** Jianxiu Hu, Qiankun Zhu, Qiong Zhang

**Affiliations:** Institute of Earthquake Protection and Disaster Mitigation, Lanzhou University of Technology, Lanzhou 730050, China

**Keywords:** footbridge, computer vision, pedestrian load, pedestrian detection, vibration comfort

## Abstract

The vibration comfort evaluation is a control standard other than strength and deflection, but the general comfort evaluation method only considers the response of the mid-span position and does not consider the difference in the vibration response of different positions at the same time. It is crucial to study how pedestrians actually feel when they walk on footbridges. The computer vision-based vibration comfort evaluation method is a novel method with advantages, such as noncontact and long-distance. In this study, a computer vision-based method was used to evaluate the global vibration comfort of footbridges under human-induced excitation. The improved Lucas–Kanade optical flow method is used for multitarget displacement identification of footbridges. Additionally, the YOLOv5 algorithm for pedestrian detection is used to obtain the position information of pedestrians on the footbridges. Then, according to the pedestrian position information, the structural responses of different pedestrian positions corresponding to time periods are extracted from the displacement responses of each point, and they are combined to obtain the structural global displacement. The global acceleration can be obtained by calculating the global displacement. The *rms* value can be calculated based on the global acceleration and compared with the standard for comfort evaluation. The global comfort evaluation method is validated by pedestrian walking experiments with different frequencies on a laboratory footbridge. The experimental results show that the computer vision-based global comfort evaluation method for footbridges is feasible and is a more specific and real-time comfort evaluation method.

## 1. Introduction

With the rapid development of urban transportation infrastructure, urban traffic flow is also increasing. At present, footbridges are commonly used in cities to ease the growing burden of urban traffic operations, making urban traffic three-dimensional and diverting people and vehicles [1,2,3]. Contrary to highways and bridges, the main load form of footbridges is pedestrian load, which is easily affected by human-induced vibration [4,5,6,7]. However, it was not until the closure of the Millennium Bridge in London that the problem of human-induced vibration attracted more and more attention from researchers [8,9,10]. At the same time, vibration comfort has also become one of the important control criteria for footbridge designs, in addition to strength and deflection deformation requirements.

Vibration comfort is an important index for evaluating the performance of footbridges. The structural response parameters required for the evaluation of comfort are generally acceleration peak value, vibration dose value (VDV) [11], or acceleration root mean square value (*rms*) [12]. Traditional testing methods are based on traditional sensors (such as accelerometers) to collect structural vibration responses [13,14,15]. The sensors have the advantages of small size, lightweight, high sampling frequency, and high precision. Chilamkuri et al. [16] introduced an intelligent sensor system for bridge structural health monitoring, using acceleration sensors to measure the vibration of the Varadhi bridge deck. This monitoring method can be applied to many infrastructures to aid in structural disaster management and recovery. However, the use of traditional sensor testing is prone to traffic jams, complicated operations, and time-consuming and labor-intensive phenomena. Therefore, noncontact laser and radar measurement methods [17,18] are increasingly used by researchers for structural displacement detection. Gichun Cha et al. [19] proposed a method to estimate the vertical displacement of a structure using light detection and ranging, enabling noncontact measurements. Laser inspection methods can cause misalignment of measurement results when measuring structures with uneven surfaces. In addition, Guan et al. [20] proposed a bridge displacement intelligent radar sensor network, which is used to measure the static and dynamic displacement of the structure, process the node data of the sensors in the network, and perform wireless transmissions. Whereas the distribution of the test positions of the radar detection method is not flexible enough, and noise will be introduced in the acquisition process, which will affect the accuracy of the measurement results. While conventional sensors are advantageous in accuracy detection, such vibration testing instruments are often expensive and difficult to set up and maintain. More importantly, when there are many locations to be measured, many devices are required to complete the detection, which is very inconvenient.

The computer vision-based vibration response acquisition method is a method of acquiring digital image information using a mobile camera device. This method has the advantages of long distance, noncontact, low cost, convenient operation, and a wide application range. Many domestic and foreign research scholars have conducted a lot of research in combination with computer vision [21,22]. Lee et al. [23] proposed a long-term displacement measurement strategy based on computer vision for self-motion compensation. The method they propose uses two cameras, the main camera uses conventional computer vision methods to measure the structural displacement, and the sub-camera is used to eliminate the measurement error of the main camera. It is used to measure the displacement of a single point in a structure. While the identification of single-target structural displacement is not sufficient for the study of structural health monitoring, Feng et al. [24] proposed a vision-sensor system for remote measurements of structural displacement. They used a template matching method for multipoint displacement measurements, whereas computer vision methods for the assessment of vibration comfort are relatively few. Dong et al. [25] proposed a noncontact footbridge vibration comfort assessment method based on computer vision. Additionally, they conducted a series of footbridge experiments with different synchronicities. Their method only studies the vibration comfort at the mid-span position of the footbridges. Nevertheless, pedestrians may feel different when walking at different positions of the footbridge.

Shahabpoor et al. [26] proposed a new concept of vibration comfort assessment based on the actual vibration level experienced by each pedestrian rather than the typical maximum vibration response at fixed points. However, this idea has only been implemented with simulation methods, and has not been implemented in the presence of real pedestrians walking. Additionally, so far, the evaluation of structural vibration comfort based on computer vision only uses the vibration response at the mid-span position to represent the pedestrian’s feelings during the whole process of walking. While computer vision-based comfort assessment is a promising assessment method with many advantages, more research is needed to consider the actual vibration levels experienced by pedestrians on this basis.

All in all, there are three difficulties that need to be overcome to realize the vibration comfort assessment based on the actual vibration level experienced by pedestrians:One of the key points for achieving the vibration comfort assessment based on the actual vibration level of pedestrians is to obtain the global displacement of the structure.This evaluation method needs to grasp the position information of pedestrians during walking.The vibration response at different positions of the structure needs to be combined with the pedestrian position to obtain the actual vibration experienced by the pedestrian at the position.

Based on these, this research proposes a computer vision-based global vibration comfort evaluation method for footbridges. This paper uses computer vision to measure the structural displacement of pedestrians at each step during walking and evaluates the comfort level of the actual vibration level of pedestrians at each step, so as to obtain the real feeling of pedestrians in the whole process of walking. This method first uses the YOLOv5 algorithm to detect pedestrians and obtain the location information of pedestrians. Secondly, the improved LK optical flow method is used to identify the structural multitarget displacement. Then, according to the pedestrian’s position, the actual structural displacement at each step of the structure is extracted from the multitarget displacement of the structure, and the global displacement response of the structure is obtained. Meanwhile, the global acceleration response is obtained by a secondary derivation of the global structural displacement. Finally, the *rms* value is calculated according to the global acceleration, so as to evaluate the global vibration comfort of the footbridges.

## 2. Global Vibration Comfort Evaluation of Footbridges Based on Computer Vision

### 2.1. The Basic Steps

The computer vision-based global vibration comfort evaluation method for footbridges proposed in this paper consists of pedestrian detection and structural multipoint displacement recognition. The so-called global vibration comfort evaluation method refers to firstly using the YOLOv5 algorithm to detect pedestrians to obtain the position of each step the pedestrian takes; secondly, using the improved LK optical flow method to detect the full structure displacement of the footbridge; then, the obtained multipoint displacement of the structure is combined with the pedestrian position information to establish the actual vibration level detection system experienced by pedestrians moving continuously on the footbridge during the walking process; the obtained actual vibration level index is used to evaluate the vibration comfort. The flowchart is shown in Figure 1.

### 2.2. Pedestrian Detection Algorithms

#### 2.2.1. YOLOv5 Algorithms

In order to obtain the real-time position information about pedestrians, the YOLOv5 algorithm needs to be used for pedestrian detection. YOLO is an end-to-end target detection model. Its core idea is to use the entire image as the input of the network, and directly return to the position and category of the bounding box in the output layer. It achieves the best balance of accuracy and speed in the current target detection algorithms. Therefore, the YOLOv5s model in the YOLOv5 algorithms is selected here to detect the position of pedestrians on the footbridge. Figure 2 is a network structure diagram of the YOLOv5s algorithm.

#### 2.2.2. Train

The training environment configuration of the YOLOv5 model is shown in Table 1.

Considering that this article only needs to identify one category of “person”, this example uses the standard dataset VOC-2007 dataset, which is a standard for measuring the ability of image classification and recognition. The dataset provides 20 categories, including 9963 pictures, a training set (5011 pictures), and a test set (4952 pictures). Among them, the pedestrian detection in this article only involves one category, namely, “person”. The person category is extracted from VOC-2007, and the YOLO configuration file is modified, recompiled, and then trained.

Training parameters: the batch size is 16, the image size is 640*640, the learning rate is 0.01, and the number of training iterations (epochs) is 500 rounds. After the training is completed, the model is saved, and the training curve is drawn. The evaluation indicators in the training process include the metric and train data line graphs. Metrics is a monitoring table for completing various data during the training process, including precision, recall rate, and loss curve. The training results are shown in Figure 3.

Training result analysis: As more and more samples are selected, the recall rate will definitely become higher and higher, and the precision will generally decline. Among them, the recall rate refers to the probability that the correct category in the sample is predicted to be correct, and the precision refers to how many of the samples whose predictions are positive are truly positive samples. As the number of training rounds increases, the precision and recall rate will increase, but after 300 rounds, the precision increases slowly, while the recall rate begins to decrease. For the loss graph, before 300 rounds of training, the loss declined, and after 300 rounds, the loss increased slightly. It shows that after 300 rounds of training, there is an overfitting phenomenon, resulting in a slight increase in loss and a decrease in recall rate. Therefore, it is best to train for approximately 300 rounds.

### 2.3. Multitarget Displacement Recognition

This section uses the improved LK optical flow method to measure the structural displacement from the footbridge vibration video. Figure 1 shows the general steps of the structure displacement detection method in this paper. First, the structure vibration video is collected, and image preprocessing is performed on it to improve the calculation speed and displacement detection accuracy. Secondly, the camera is calibrated and the conversion relationship between the image coordinate system and the real coordinate system is calculated. Then, the improved LK optical flow method is used to select the detection feature points from the image for target tracking. Finally, by comparing the position changes of the feature points in each frame of the image, the displacement in the image coordinate system is calculated, and the final displacement in the actual coordinate system is obtained by combining the camera calibration results.

The method in this paper selects the bilateral filtering method to preprocess the collected images, which can not only effectively remove noise, but also preserve the image edges.

In multitarget displacement identification, a portable SLR camera is placed in front of the footbridge, and the target detection and tracking are carried out on multiple target points of the whole bridge at the same time. Since only the longitudinal displacement needs to be measured, the multitarget displacement identification part adopts a simplified camera calibration method, that is, the scale factor method [27]. The scale factor method equation used is shown in Equations (1) and (2). Among them, Equation (1) is the calculation formula of the scale factor *s* when the optical axis of the camera is perpendicular to the structural plane, and Equation (2) is the calculation formula of the scale factor *s* when the optical axis of the camera is at an angle with the structural plane. Since the optical axis of the camera is always perpendicular to the structure plane during the shooting process, all the target points are calculated using Equation (1) for the scale factor.
(1)s=Dd
(2)s=Dd·cosα

Among them, *D* is the physical unit size of the selected target, *d* is the pixel unit size corresponding to the selected target, and *α* is the angle between the optical axis of the camera and the normal of the structure plane.

In this paper, the method used for structural displacement detection is the improved LK optical flow method [28]. The Lucas–Kanade optical flow method was proposed by Bruce D. Lucas and Takeo Kanade. It assumes that the optical flow is a constant in the field of pixels, and then uses the least squares method to solve the basic optical flow equation for all pixels in the field. The premise of the LK method is that the moving distance of the target in the image between two consecutive frames before and after is not large, and there is approximate motion consistency around the fixed point.

In this paper, when measuring the structural displacement under human-induced vibration, the bridge vibration may be severe when pedestrians pass through the bridge. At this time, the second assumption may not be true, resulting in an error in the algorithm. The improved LK optical flow method mentioned in this article combines the SIFT feature point matching with the LK optical flow method. First, feature points are extracted in the first frame (previous frame), and then feature points are matched in the second frame (current frame). Then we use optical flow to solve the corresponding relationship between the feature points of the two frames. When pedestrians pass through the footbridge, the contact between the pedestrian’s feet and the bridge deck may be recognized as a corner point. Since the SIFT feature points have good stability under environmental disturbances, such as illumination, noise, viewing angle, zoom, and rotation, they can overcome the original shortcomings of the LK optical flow method, so the SIFT feature point matching method is combined with the optical flow method.

The improved LK optical flow method can obtain the image pixel displacement of multiple positions of the footbridge structure during the same vibration process, and it is necessary to further convert the pixel displacement into the real displacement. Using the scale factor obtained by the multitarget camera calibration method, the pixel displacement of each point is converted into the real physical displacement, and the conversion equation is shown in Equation (3).
(3)D=s⋅d

### 2.4. Global Comfort Assessment

#### 2.4.1. Global Acceleration Extraction

The computer vision-based global vibration comfort evaluation method for footbridges first obtains the real-time position of pedestrians on the footbridge through a pedestrian detection algorithm based on deep learning. Then, the vibration response of the structure is obtained with the multiobjective structural displacement identification method. The structural displacement responses at each step position are extracted and combined to obtain a set of structural displacement curves, that is, the global displacement response of the footbridge. Additionally, the global displacement response of the footbridge is obtained with the acceleration calculation method described above. The *rms* value of the global acceleration is then calculated, and the vibration comfort is assessed against the vibration limits. Figure 4 depicts the method for extracting the global displacement of the footbridge mentioned in this method. As shown in Figure 5, picture a is the pedestrian position, and picture b is the structural displacement response curve at point *i*.

(1)When the pedestrian walks to the measuring point *i*, according to the pedestrian coordinate information obtained by the pedestrian detection, the moment when the pedestrian is at the middle position between the measuring point *i* − 1 and the measuring point *i* can be obtained as *t_i_*_1_. The time at the middle position between the measuring point *i* and the measuring point *i* + 1 is *t_i_*_2_.(2)Corresponding to the part from *t_i_*_1_ to *t_i_*_2_ of the structural displacement time-history curve intercepted to the measuring point *i*, the real displacement response of the pedestrian at the measuring point *i* is obtained.(3)The respective real displacement responses are intercepted from the displacement responses of n measuring points and combined into a time sequence, which is the global displacement curve of the footbridge described in this paper.

This paper describes a computer vision-based global vibration comfort assessment method for footbridges. Therefore, after obtaining the structural global displacement, the structural global displacement data needs to be converted into the structural global acceleration data through the second-order derivative. In this way, the comfort evaluation index related to the global acceleration is calculated.

#### 2.4.2. Comfort Assessment Specifications

Most of the evaluation indicators used in previous comfort evaluation methods are acceleration peaks, but the computer vision-based footbridge global vibration comfort evaluation method mentioned in this paper is closely related to the walking position of pedestrians. Therefore, considering the influence of time on the evaluation index, this paper selects *rms* (root mean square value of acceleration) as the comfort evaluation index of this method. This section states the ISO 2631-1 (1997) standard for evaluating the vibration comfort of footbridges.

This paper adopts the basic evaluation method in the ISO2631-1 (1997) specification. For the so-called basic evaluation method, the vibration intensity index is the root mean square acceleration *rms* after frequency weighting, which is calculated according to Equation (4):(4)$$rms={\left[\frac{1}{T}\int_{0}^{T}a_{w}^{2}\left(t\right)dt\right]}^{\frac{1}{2}}$$

Among them, *rms* is the weighted acceleration *rms* value, in m/s^2^ or rad/s^2^; *aw*(*t*) is the weighted acceleration (including translation and rotation) as a function of time, in m/s^2^ or rad/s^2^; *T* is the vibration duration, the unit is s.

The vibration limits used in this standard include perception limits and comfort limits.

(1)Perception limit: Fifty percent of alert and robust people have a detection limit of 0.015 m/s^2^ (peak).(2)Comfort limits: As shown in Table 2.

As shown in Table 2, when pedestrians pass, the comfort assessment of the footbridge is based on the above vibration limits. Therefore, the displacement time history is first obtained by the improved LK optical flow method described in Section 2.3, and then the acceleration time history is obtained by calculating the displacement time history. The acceleration root mean square value is calculated based on the acceleration data, and the root mean square value is compared with the vibration limit to evaluate whether the vibration comfort meets the specification requirements.

## 3. Laboratory Realization

### 3.1. Experimental Overview

In order to verify the effectiveness of the method described in this article for evaluating the overall structural comfort of a footbridge using computer vision, a series of experiments were carried out on a single-span simply supported footbridge model in the laboratory. The bridge is a reduced model of a medium-sized actual structure. The total mass of the bridge deck is 1580 Kg, the deck length is 10 m, the width is 1.6 m, and the calculated span is 9.8 m, as shown in Figure 5. The bridge deck is laid with five tempered glass panels, each with a length of 2 m and a width of 1.6 m. Each layer of tempered glass is 10 mm thick, and the total thickness is approximately 22 mm.

To synchronously measure the position of the pedestrian and the vibration of each point of the footbridge structure, a Canon SLR camera was installed at approximately 4 m from the middle of the bridge span to track the walking process of the pedestrian. The SLR camera can be connected to the mobile phone’s Bluetooth for remote noncontact control to avoid errors caused by camera shake due to contact manipulation. The video resolution is 1920 × 1080, and the frame rate is 50 fps per second. At the same time, a laser displacement meter and a 941B acceleration sensor were installed under the bridge to verify the accuracy of the visual measurement of displacement and acceleration. The laser displacement meter model used is the Banner 250U, and the sampling frequency was set to 50 Hz.

The layout of the test site is shown in Figure 5.

### 3.2. Experimental Data Analysis

#### 3.2.1. Identification Method Verification

In this section, a test subject with a weight of 81 Kg is taken to analyze the working conditions of the footbridge at a random cadence. The displacement time history and acceleration time history of a point are measured by a laser displacement meter and an acceleration sensor, respectively. As shown in the figure, Figure 6 shows the time-history comparison chart and the frequency-domain comparison chart of the mid-span displacement of the footbridge measured with a visual recognition device and a laser displacement meter when a tester weighing 81 Kg passes the footbridge at a random cadence. Figure 7 is a time-domain comparison diagram and a frequency-domain comparison diagram of the acceleration results calculated from the visual recognition displacement result and the results collected with the acceleration sensor in this process.

As shown in Figure 6 the structural displacement curves obtained by the two methods are basically consistent, the peak displacement measured by the laser displacement meter is 4.438 mm, and the peak value of the structural displacement obtained by the visual recognition method is 4.336 mm, and the comparison error is 2.298%; under the domain analysis, a peak appears at the fundamental frequency of the structure at 4.3 Hz, the visual recognition result is 0.1518, the laser displacement meter recognition result is 0.1558, and the comparison error is 2.567%. Therefore, the visual recognition method of the portable SLR camera can accurately measure the structural vibration response under human-induced excitation.

As shown in Figure 7, the peak value of the acceleration time-history curve converted from the visual recognition displacement results is 0.2093 m/s^2^, and the peak value of the acceleration measured by the acceleration sensor is 0.2117 m/s^2^. The difference between the two is 0.0024 m/s^2^, and the comparison error is 1.133%; in the frequency-domain comparison, both have peaks at 1/2 the fundamental frequency of the structure. It can be seen that this conversion method has a small error and can replace the test results of traditional acceleration sensors.

#### 3.2.2. Pedestrian Detection

We used the pedestrian detection algorithm mentioned in Section 2.2 to identify pedestrians on the footbridge in the laboratory; the pixel position information of the pedestrians can be obtained. Firstly, the camera calibration needed to be carried out, and the scale factor was obtained according to the ratio of the recognized size of the target in the image to the real size. Then, we use the YOLOv5 algorithm to detect pedestrians, and track pedestrians to obtain real-time location information about pedestrians. Using the scale factor we obtained from the camera calibration, the pixel coordinates of the identified pedestrians were converted into real coordinate information in reality, so as to realize the position estimation of pedestrians on the bridge.

The pedestrian detection results obtained in the experiment are shown in Figure 8. The pedestrian position coordinates output by the experiment are pixel coordinates. The graph obtained by plotting the pixel coordinates is shown in Figure 9.

#### 3.2.3. Global Result Recognition

In order to obtain the global vibration characteristics of the footbridge, the aforementioned improved LK optical flow method was used to identify the multipoint displacement of the footbridge. We let a pedestrian walk on the footbridge at a random pace, and set the maximum number of identification points to 10, according to the span of the laboratory footbridge. First, in the process of multipoint displacement identification, the displacement of a certain point is identified and compared with the results collected by the displacement meter, so as to verify the accuracy of the multipoint identification method. Then, the displacement time history curve of 10 points is obtained.

Figure 10 is a time-domain comparison diagram and a frequency-domain comparison diagram of the displacement across the midpoint during multipoint identification. Figure 11 shows the displacement time-history curve of 10 points from the bridgehead to the bridge tail.

The time area is divided by proposing the pedestrian walking position, and the structural displacement of each time area is extracted from the multipoint displacement recognition results. The global displacement of the pedestrian bridge is then combined during walking, as shown in Figure 12. Through the calculation, the global acceleration time-history curve of the footbridge is obtained, as shown in Figure 13.

We used the vibration comfort evaluation method mentioned in Section 2.4 to evaluate the instantaneous acceleration results of the footbridge and the acceleration results of the mid-span position, respectively, that is, we calculated the *rms* values of the two kinds of acceleration time history with Equation (4) for comparison. The calculation results are as follows: the instantaneous root mean square value of the footbridge acceleration is 0.05154 m/s^2^, and the root mean square value of the acceleration at the mid-span position is 0.04351 m/s^2^. Compared with the vibration limit table, the comfort evaluation results of pedestrians show that they can perceive the bridge vibration, but not feel uncomfortable. However, according to the different acceleration root mean square values, the vibration comfort under the influence of a time factor and the pedestrian’s position is not completely consistent with the typical maximum response of the mid-span position only. Therefore, it is not possible to simply use the typical maximum response at the mid-span position of the footbridge for comfort assessment.

#### 3.2.4. Experimental Results

As mentioned above, in this experiment, five pedestrians wore sports shoes with hard soles and walked from the bridgehead to the end of the bridge under three working conditions of fixed step frequency 1.8 Hz, 2.0 Hz and 2.2 Hz. The detailed parameters of the five testers are shown in Table 3.

Five pedestrians walked from the bridgehead to the end of the bridge under three working conditions of fixed step frequency of 1.8 Hz, 2.0 Hz, and 2.2 Hz, and analyzed various working conditions by using the abovementioned method of extracting the real-time acceleration of the footbridge in the whole area. The global acceleration of the footbridge under this working condition is shown in Figure 14. We calculated the *rms* values of the two acceleration time histories with Equation (4), respectively, for comparison, and obtained the comparison results of the *rms* values of the two acceleration time histories under 15 working conditions, as shown in Figure 15. As shown in Figure 15, the value of the *rms* calculated by the acceleration time history of the mid-span position is larger than the value of the *rms* obtained from the global acceleration of the footbridge. Even in the case of pedestrian 5 walking at 2.2 Hz, the *rms* value of the mid-span position is relatively small, which further illustrates that the traditional method of comfort assessment based on the typical maximum response of the mid-span position is not representative. It should be combined with the pedestrian’s position on the footbridge to evaluate the comfort of pedestrians at all times.

## 4. Conclusions and Discussion

On the basis of previous research, in order to more accurately evaluate the vibration comfort of footbridges, this paper proposes a computer vision-based global vibration comfort evaluation method for footbridges that considers the influence of pedestrian position on the vibration comfort of the footbridge. This method is different from the previous footbridge vibration comfort evaluation method, which uses the typical maximum response at the mid-span position and that represents the full-bridge response. Instead, this method extracts the real responses of pedestrians at each step from the multipoint displacement identification results of the footbridge according to the pedestrian’s positions, and then combines the real responses at each point to obtain the global response of the footbridge. According to the experimental data listed in this paper, the following conclusions were drawn:

(1)Through testing the footbridge in the laboratory, the error of the vision displacement result compared with the displacement response collected by the displacement meter is 2.298%, the error is small, and the accuracy is high.(2)The YOLOv5 algorithm can quickly and accurately perform target detection, and can obtain the position information of pedestrians in the process of walking. It is a feasible pedestrian detection algorithm.(3)By comparing the global acceleration *rms* value of a single pedestrian walking on the footbridge at three frequencies of 1.8 Hz, 2.0 Hz, and 2.2 Hz with the acceleration *rms* value at the mid-span position of the footbridge, it is found that the *rms* value calculated by the acceleration time history of the mid-span position is generally larger than the *rms* value obtained by the global acceleration of the footbridge. However, there are also cases where the acceleration *rms* value at the mid-span position is small. It shows that the traditional method of evaluating the comfort with the typical maximum response of the mid-span position is not accurate, but should be combined with the pedestrian’s position on the footbridge to evaluate the comfort of pedestrians at every moment.

Due to technical limitations, this method can only extract the global displacement of the footbridge according to the measured multipoint displacement results of the structure and the pedestrian position information after pedestrian detection, and calculate the global acceleration and its *rms* value, but cannot obtain real-time information about the pedestrian’s position at each step. In order to realize the real-time identification of the structural response where pedestrians go, it is necessary to further improve the model and algorithm, so as to realize the global vibration comfort evaluation faster. Further research can verify that the vibration comfort evaluation method for the entire area of the footbridge mentioned in this article can be verified in the case of pedestrians walking. We can judge whether this method of comfort evaluation still has advantages in the case of pedestrians walking, and apply this method to engineering practices.

## Figures and Tables

**Figure 1 sensors-22-07077-f001:**
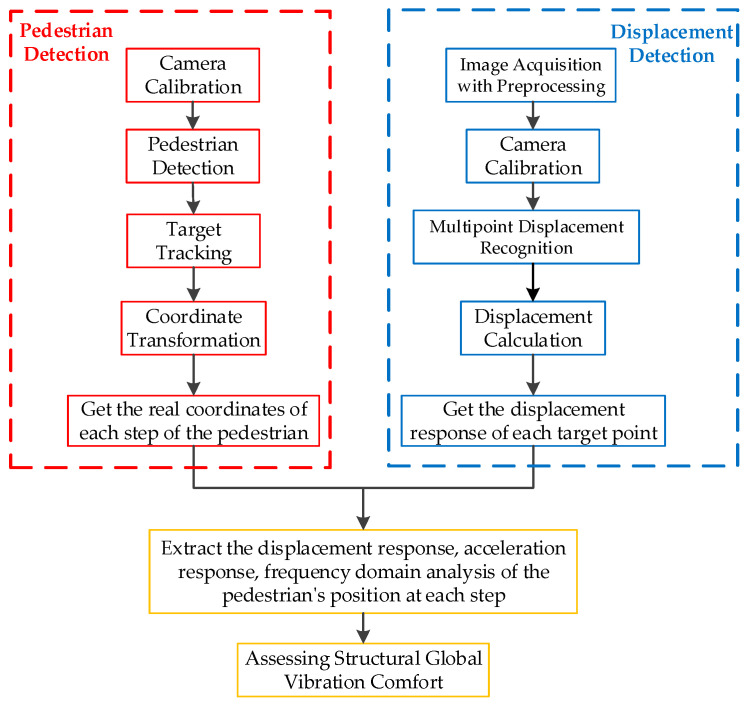
Flow chart.

**Figure 2 sensors-22-07077-f002:**
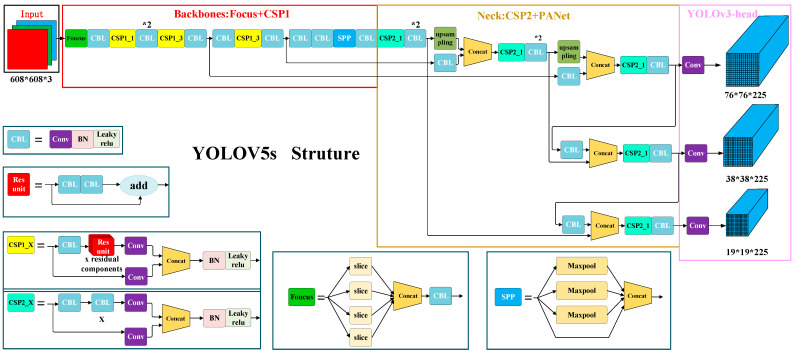
YOLOv5s network structure diagram. Internet source.

**Figure 3 sensors-22-07077-f003:**
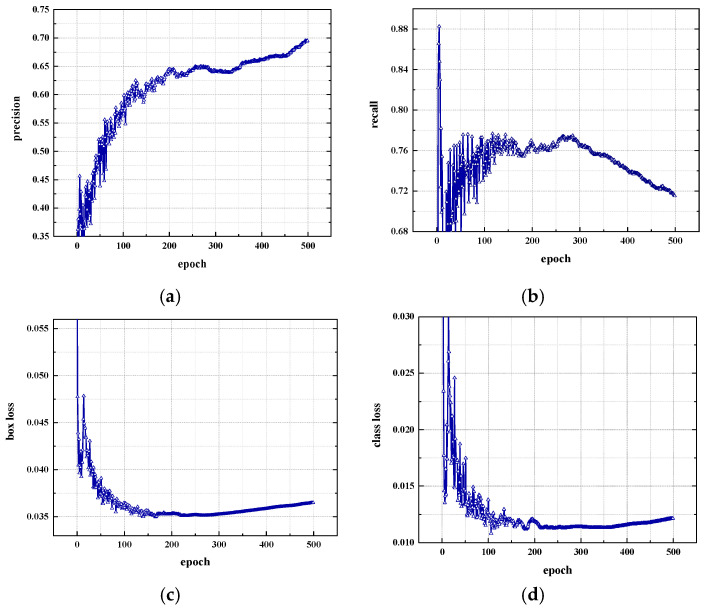
Training result. (**a**) precision; (**b**) recall; (**c**) box loss; (**d**) class loss.

**Figure 4 sensors-22-07077-f004:**
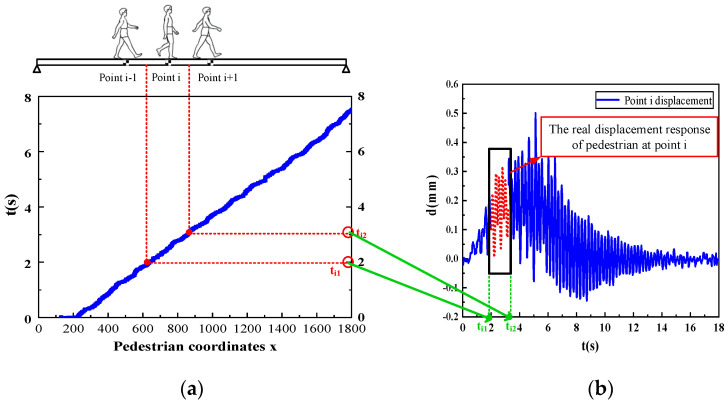
Global displacement extraction method. (**a**) Pedestrian position; (**b**) displacement response of point *i*.

**Figure 5 sensors-22-07077-f005:**
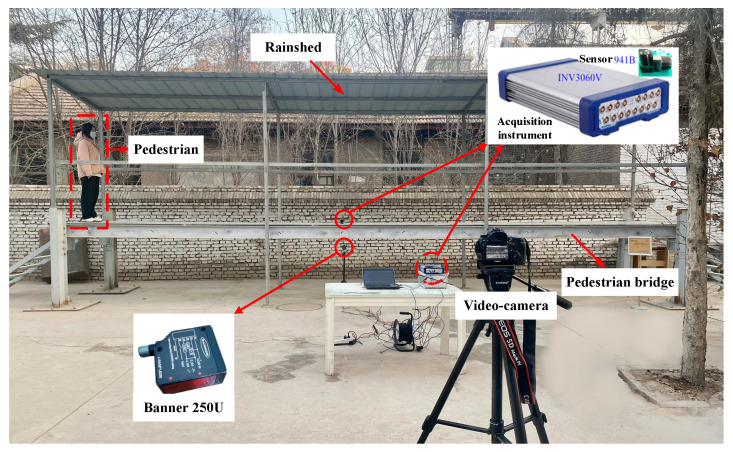
Test site layout.

**Figure 6 sensors-22-07077-f006:**
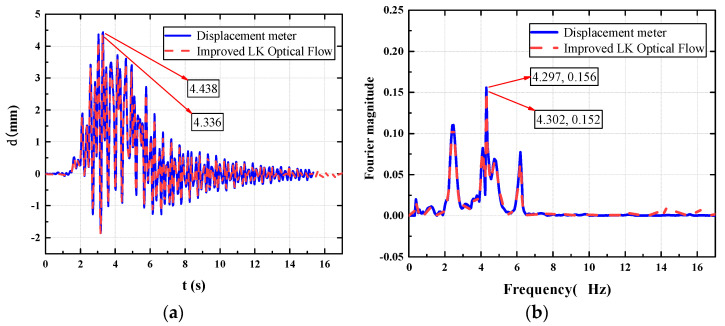
Displacement comparison chart. (**a**) Displacement time-history comparison chart; (**b**) displacement frequency-domain comparison chart.

**Figure 7 sensors-22-07077-f007:**
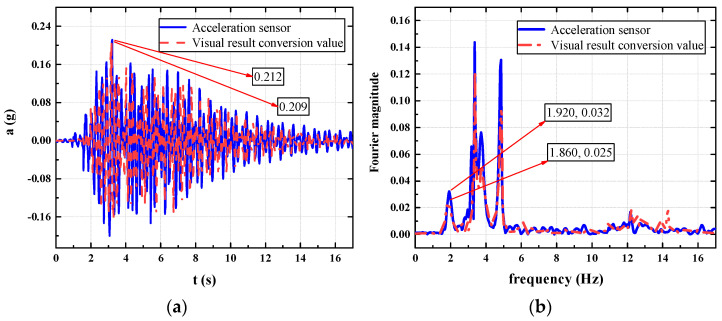
Acceleration comparison chart. (**a**) Acceleration time-domain comparison chart; (**b**) acceleration frequency-domain comparison chart.

**Figure 8 sensors-22-07077-f008:**
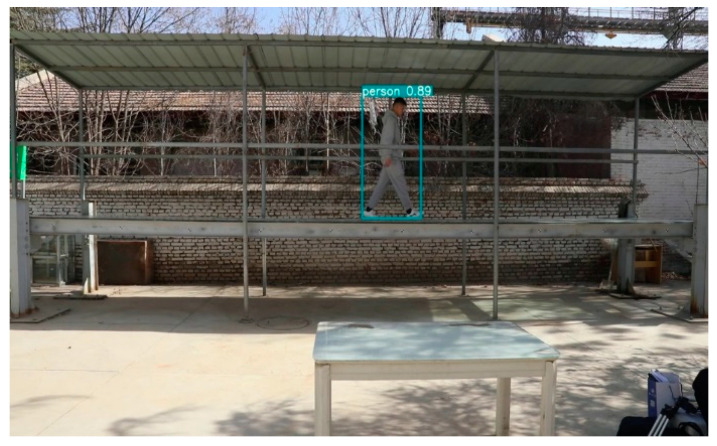
Pedestrian detection result graph.

**Figure 9 sensors-22-07077-f009:**
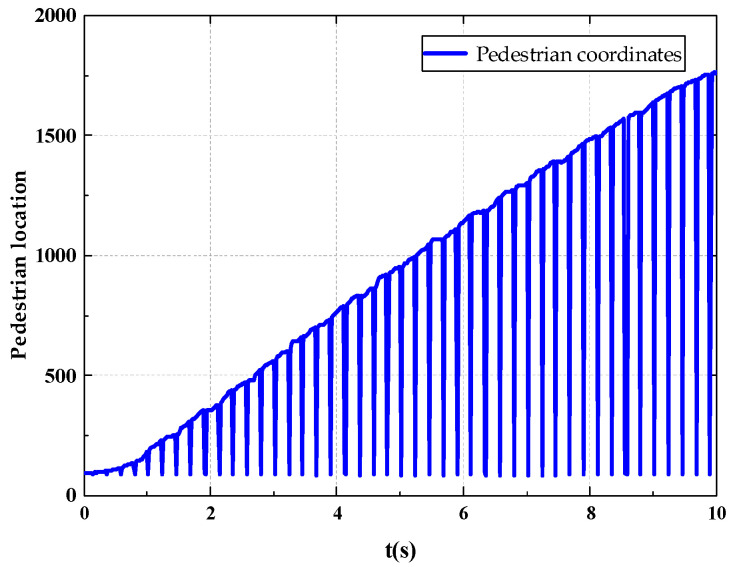
Pedestrian Coordinate Chart.

**Figure 10 sensors-22-07077-f010:**
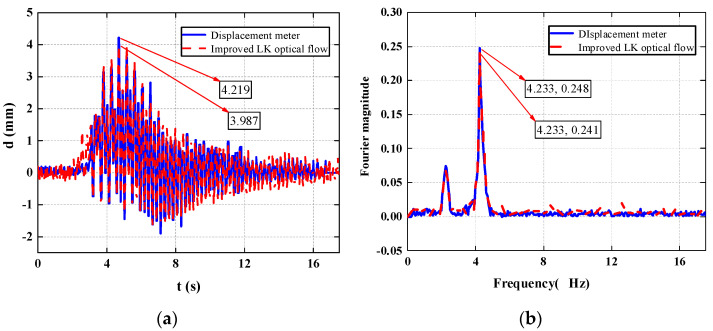
Mid-span displacement comparison chart. (**a**) Displacement time-history comparison chart; (**b**) displacement frequency-domain comparison chart.

**Figure 11 sensors-22-07077-f011:**
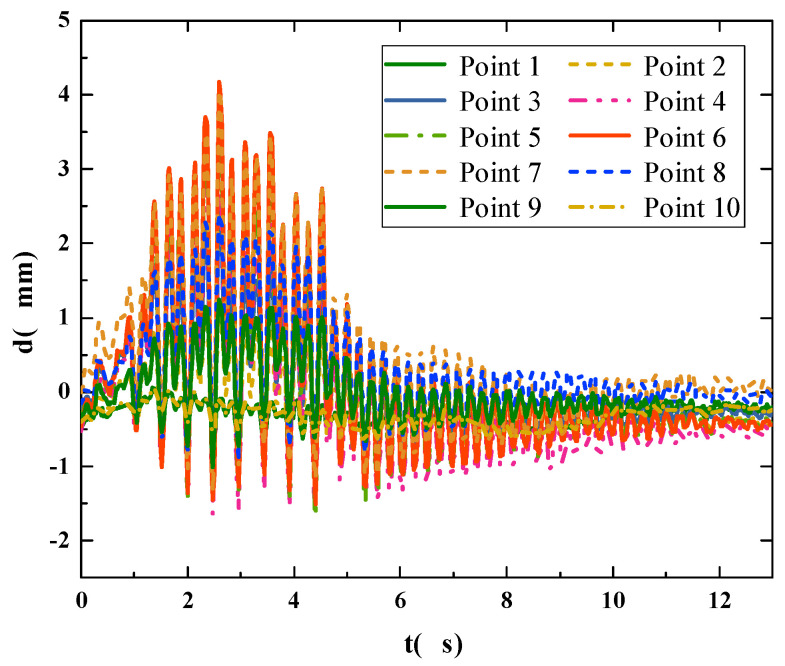
Multipoint displacement recognition results.

**Figure 12 sensors-22-07077-f012:**
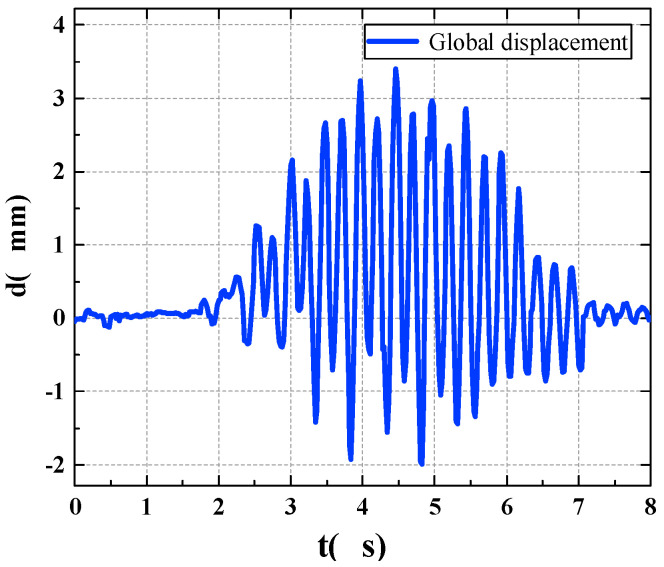
Global displacement of footbridge.

**Figure 13 sensors-22-07077-f013:**
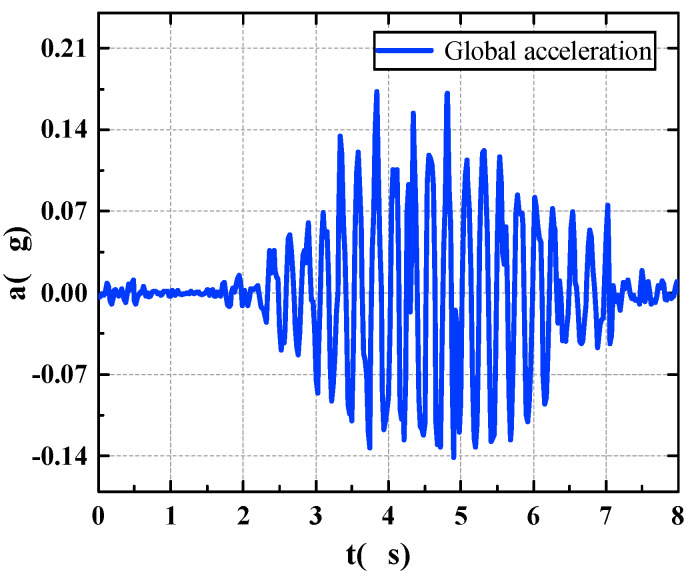
Global acceleration of footbridge.

**Figure 14 sensors-22-07077-f014:**
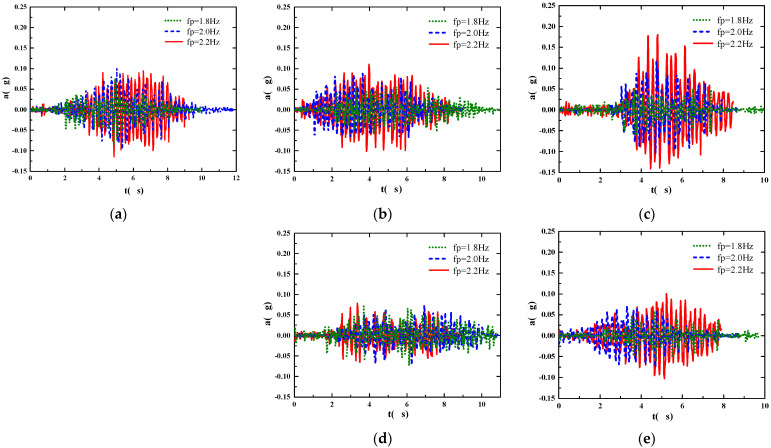
Global acceleration of the footbridge under 15 working conditions. (**a**) Global acceleration of pedestrian 1 at three frequencies; (**b**) global acceleration of pedestrian 2 at three frequencies; (**c**) global acceleration of pedestrian 3 at three frequencies; (**d**) global acceleration of pedestrian 4 at three frequencies; (**e**) global acceleration of pedestrian 5 at three frequencies.

**Figure 15 sensors-22-07077-f015:**
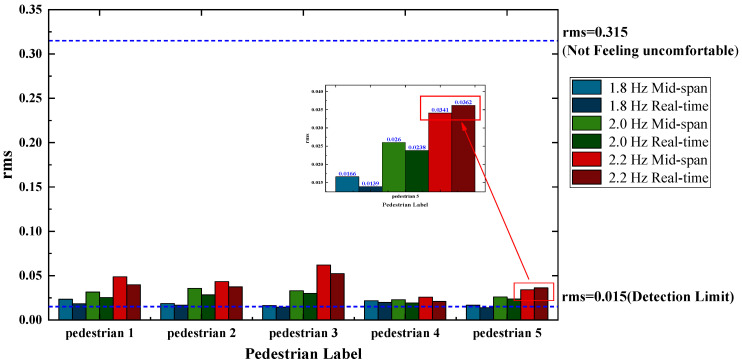
Global *rms* of the footbridge under 15 working conditions.

**Table 1 sensors-22-07077-t001:** Experimental environment configuration.

Environment	Version
Computer System	Windows10 LTSC2019
CPU	Inter Core™ i7-9700F
GPU	GeForce RTX 2070 SUPER
Hardware Acceleration	CUDA10.2; CUDNN7.6

**Table 2 sensors-22-07077-t002:** Vibration limit (ISO 2631-1(1997)).

*rms*	Label
<0.315 m/s^2^	Not Feeling uncomfortable
0.315–0.63 m/s^2^	A little uncomfortable
0.5–1 m/s^2^	Quite uncomfortable
0.8–1.6 m/s^2^	Uncomfortable
1.25–2.5 m/s^2^	Very uncomfortable
>2 m/s^2^	Extremely uncomfortable

**Table 3 sensors-22-07077-t003:** The experimental pedestrian parameters.

Pedestrian Label	Gender	Weight (Kg)	High (m)	Shoulder Width (cm)
1	Female	54	1.62	38
2	Male	89	1.82	50
3	Male	65	1.80	41
4	Male	80	1.80	42
5	Female	53	1.64	40

## Data Availability

The data that support the findings of this study are available from the corresponding author upon reasonable request.

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
