# Peer review of "Global Vibration Comfort Evaluation of Footbridges Based on Computer Vision"

_sensors, 2022, doi:10.3390/s22187077_

Round 1

Reviewer 1 Report

In this paper, the YOLOv5 algorithm and the improved LK optical flow method are used to extract the actual structural displacement synchronized with the pedestrian movement from the multi-objective displacement of the structure, and then evaluate the global vibration comfort of the pedestrian bridge. There are some comments as follows:

1. The concept of this paper has been proposed by Shahapur et al. The research in this paper is only an experimental realization of the proposed concept. The innovation points are less clear, and the innovation points of this paper need to be further clarified.

2. For the difficulty 1 proposed in this paper in lines 95-96, this difficulty is only for the traditional sensor measurement. For the multi-point displacement measurement using computer vision technology introduced in this paper, it has been solved before, and no sensor is needed, which is not a difficulty. This can be changed to: One of the key points of realizing the vibration comfort assessment based on the actual vibration level of pedestrians is to obtain the global displacement of the structure.

3. For the analysis of the training results described in lines 158-166 of this article: As mentioned above, the samples for program training in this article are only "people". Why does the text point out the impact of sample size on recall and precision? What is the purpose? Moreover, in Figure 3, the changing trend of recall rate and accuracy rate with increasing number of training rounds has not been analyzed. Here, the optimal number of training rounds is obtained by judging the change trend of the loss parameter with the increase of the number of training rounds, but the meaning of the parameter loss is not explained. Why can the judgment be made by the loss parameter? What is the relationship between the loss parameter and the recall rate and accuracy rate?

4. Lines 186-188 in the text point out that the scale factor method is used to convert the pixel coordinate system to the real coordinate system. Is the influence of camera distortion on the coordinate system conversion relationship at the edge of the image considered?

5. The D and d variables in Equation 3 and Equations 1 and 2 are used repeatedly, which is easy to cause confusion. Should a distinction be made between the unit size used for calibration and the actual test unit size used for measurement?

6. It can be seen from the method described in Section 2.4.1 of this paper that the research in this paper does not achieve the global acceleration extraction in the true sense. That is to say, the displacement and acceleration extracted by this study are not completely in accordance with the real-time motion trajectory of the pedestrian, but the structural displacement and acceleration when the pedestrian walks to the feature point are extracted through the feature points calibrated in advance. It is an approximate global displacement response and global acceleration calculation method. The accuracy of this calculation method is largely determined by the number of feature points arranged.

7. In line 291 of the text, "as shown in Figure 7" is quoted incorrectly, here it should be Figure 6.

8. In line 349 in the text: According to the ordinate values in Figure 10, it can be analyzed that the figure should be a moving figure of pedestrians in pixel coordinates, not the figure obtained in the real coordinates described in the text.

9. For lines 359-361 in the text: only one point is compared to verify the accuracy of all points. Are there regional contingencies? Are there too few control groups?

10. As described in Section 3 of this paper: the experiments designed in this study only carried out the following two sets of comparative experiments: (1) The displacement and acceleration time-history curves of a single point obtained by visual recognition were compared with the results measured by the instrument. (2) The rms value obtained from the global acceleration is compared with the rms value obtained from the midspan position acceleration. But there is no comparative test to prove that the global acceleration obtained by the method described in this paper is correct and can be used. This test does not use currently accepted methods or instruments to measure the widely accepted global acceleration for comparison of results.

11. Modal parameters also affect vibration comfort, which should be reviewed more and introduced briefly. For the reviews on this topic refer to: Closely spaced modes identification through modified frequency domain decomposition, Measurement, 2018, 128:388-392; Frequency identification of practical bridges through higher order spectrum, Journal of Aerospace Engineering-ASCE, 2018, 31(3): 04018018; Mode identification by eigensystem realization algorithm through virtual frequency response function, Structural Control and Health Monitoring, 2019, 26(10): e2429

Reviewer 2 Report

Computer vision of pedestrian movement on a footbridge is a good idea, but dynamic image processing parameters are difficult to evaluate with precision. therefore, how far does the precision ratio in this experiment

What is the algorithm YOLOv5? Please describe how the algorithm in this study works and how it helps to detect global displacement. You have explained this in a later section; however, please provide a glimpse in the introduction.

Figure 1 is an excellent flowchart that depicts the entire research process, but it would be even better if it were coloured. Consider that if possible? 

could you please explain what the LK optical flow method is and how it operates?

Vibration evaluation is dependent on pedestrian movements in a footbridge, and you have successfully processed images based on this research; how about irregular movements, such as running, or the transfer of large equipment, etc., and whether the same method is applicable? If so, how extensive would it be?

figure 2 "YOLOv5s network structure diagram" is unclear; please improve the image quality.

Experimentally, the height of the footbridge from ground level appears to be quite low. Consequently, the deviation in pedestrian behaviour will be anticipated and foreseeable. how the same techniques will be applicable if the ground level is deeper

You mentioned in the conclusion section that, due to technical constraints, these methods can only extract global displacement. Please describe the technical constraints you encountered during your research.

Reviewer 3 Report

The paper

“Global vibration comfort evaluation of footbridges based on computer vision”,

by Jianxiu Hu et al.,

presents a remote sensing, computer vision-based strategy for the vibration comfort evaluation of slender footbridges. This approach jointly uses LK optical flow method for multi-target object tracking and YOLOv5 for pedestrian identification.

The aim of the research is clearly stated and the concept is of great interest to researchers in the field of Bridge Monitoring.

However, there are some points, regarding both the paper’s content and format, that need to be addressed to grant full acceptance. In more detail:

 1.      In the abstract, since it has not been introduced yet, it should be explicitly written Lucas–Kanade optical flow rather than LK optical flow.

2.      The dynamic response of slender footbridges is strongly influenced by their actual slenderness. In this context, it would be useful to compare the mass per unit of length (here, 158 kg/m lin) of the laboratory-simulated structures to actual, real-life and real-size footbridges.

3.      In their Introduction, the Authors mention that the comfort can be evaluated in terms of vibration dose value (VDV) or acceleration root mean square value (rms). However, in the rest of the paper, only this second index is actually utilised. For completeness, it would be useful to report the same results in terms of VDV as well.

4.      From the abstract, the proposed approach is (correctly) intended for long-distance non-contact monitoring. However, in real-life applications, the camera would be likely farther away from the target structure (here, it was just 4 m). This could make the displacements too small (in terms of pixels) to be accurately tracked. In this regard, Phase-Based Motion Magnification (PBMM) was proposed for such circumstances. This similar computer vision-based approach can be recalled in the state-of-the-art review (ref 21 is already included in the text); in particular, https://doi.org/10.1111/str.12336 discusses its application for vibration-based structural assessment.

5.      There is a clear issue with Figure 10, as it seems that the identified coordinate defaults to zero at regular times.

6.      Figure 11, is it really necessary to indicate the frequency resolution up to the fifth decimal digit? This is not very realistic.

7.      In the subtitle, Lab realization should be Laboratory realization.

8.      Table 3, third column: for consistency, use the same number of decimal digits (none or one, as for the second-to-last row).

9.      The blank space between the numeric value and the following measurement unit is often missing (e.g. 0.315m/s2 should be 0.315 m/s2). This applies also to some figures (e.g. the legend in Figure 16).

10.   The notation for the measurement units on the x and y axes is not consistent for all figures - somewhere /s and /mm, somewhere [s] and [mm], somewhere else (s). In certain cases, such as the x-axis of Figure 5.a, the measurement unit is (wrongly) omitted.

11.   Please double-check carefully throughout the text for grammar mistakes and typos. E.g. page 2 line 53, “While, laser inspection”.

12.   In the label of the y-axis of Figure 3.d, it should be “class” and not “cls”.

13.   Is Figure 2 an original contribution from the authors? Otherwise, if retrieved from the already published scientific literature, the original source should be reported in the figure caption.

14.   Table 2: the original source (ISO 2631-1 (1997)) should be reported in the caption.

15.   Figure 4 is redundant since it does not add any further information with respect to Figure 1. Thus, it is suggested to remove it.

Round 2

Reviewer 2 Report

The author has made all the corrections as per the disscution so the article can be published in its present form.

Reviewer 3 Report

the authors addressed all the comments raised by this reviewer. The manuscript can be accepted in its current form after proper grammar checking and proofreading.